# Mortality and evolution between community and hospital-acquired COVID-AKI

**Jonathan S. Chávez-Íñiguez**[1,2]*, **José H. Cano-Cervantes**[3,4], **Pablo Maggiani-Aguilera**[1,2], **Natashia Lavelle-Góngora**[1,2], **Josué Marcial-Meza**[3,4], **Estefanía P. Camacho-Murillo**[3,4], **Cynthia Moreno-González**[1,2], **Jarumi A. Tanaka-Gutiérrez**[1,2], **Ana P. Villa Zaragoza**[1,2], **Karla E. Rincón-Souza**[1,2], **Sandra Muñoz-López**[3,4], **Olivia Montoya-Montoya**[4], **Guillermo Navarro-Blackaller**[1,2], **Aczel Sánchez-Cedillo**[3,4], **Luis E. Morales-Buenrostro**[4,5], **Guillermo García-García**[1,2]

1 Renal Transplant Unit, Nephrology Department, Civil Hospital of Guadalajara Fray Antonio Alcalde, Guadalajara, México, 2 Nephrology Department, University of Guadalajara Health Sciences Center, Guadalajara, Mexico, 3 Renal Transplant Unit, Nephrology Department, National Medical Center ISSSTE 20 de Noviembre, Mexico City, Mexico, 4 Nephrology Department, National Autonomous University of Mexico, Mexico City, Mexico, 5 Renal Transplant Unit, Nephrology Department, National Institute of Medical Sciences and Nutrition Salvador Zubirán, Mexico City, Mexico

* jonarchi_10@hotmail.com

**Data Availability Statement:** All relevant data are within the manuscript and its Supporting Information files.

## Abstract

### Background

Acute kidney injury (AKI) is associated with poor outcomes in COVID patients. Differences between hospital-acquired (HA-AKI) and community-acquired AKI (CA-AKI) are not well established.

### Methods

Prospective, observational cohort study. We included 877 patients hospitalized with COVID diagnosis at two third-level hospitals in Mexico. Primary outcome was all-cause mortality at 28 days compared between COVID patients with CA-AKI and HA-AKI. Secondary outcomes included the need for KRT, and risk factors associated with the development of CA-AKI and HA-AKI.

### Results

A total of 377 patients (33.7%) developed AKI. CA-AKI occurred in 202 patients (59.9%) and HA-AKI occurred in 135 (40.1%). Patients with CA-AKI had more significant comorbidities, including diabetes (52.4% vs 38.5%), hypertension (58.4% vs 39.2%), CKD (30.1% vs 14.8%), and COPD (5.9% vs 1.4%), than those with HA-AKI. Patients' survival without AKI was 87.1%, with CA-AKI it was 75.4%, and with HA-AKI it was 69.6%, log-rank test p < 0.001. Only age > 60 years (OR 1.12, 95% CI 1.06–1.18, p <0.001), COVID severity (OR 1.09, 95% CI 1.03–1.16, p = 0.002), the need in mechanical lung ventilation (OR 1.67, 95% CI 1.56–1.78, p <0.001), and HA-AKI stage 3 (OR 1.16, 95% CI 1.05–1.29, p = 0.003) had a significant increase in mortality. The presence of CKD (OR 1.48, 95% CI 1.391.56, p < 0.001), serum lymphocytes < 1000 µL (OR 1.03, 95% CI 1.00–1.07, p = 0.03), the need in

**Funding:** The author(s) received no specific funding for this work.

**Competing interests:** The authors have declared that no competing interests exist.

mechanical lung ventilation (OR 1.06, 95% CI 1.02–1.11, p = 0.003), and CA-AKI stage 3 (OR 1.37, 95% CI 1.29–1.46, p < 0.001) were the only variables associated with a KRT start.

## Conclusions

We found that COVID patients who are complicated by CA-AKI have more comorbidities and worse biochemical parameters at the time of hospitalization than HA-AKI patients, but despite these differences, their probability of dying is similar.

## Introduction

COVID-19 has been a unique challenge to the field of nephrology. Acute kidney injury (AKI) presents in 4.5% of cases [1] and up to 78% of patients in the intensive care unit [2]. AKI is associated with poor outcomes [3], and it has been reported that AKI has the highest risk of mortality in hospitalized patients [4]. Mexico has been one of the most adversely affected countries in the world by the pandemic [5]. Up to 37% of COVID-19 patients present to the hospital with AKI, that is, they acquire it in the community (CA-AKI) [6], an event that can be explained by the fact that COVID patients develop such symptoms as fever, cough, anorexia, diarrhea, and fatigue several days before going to the hospital [7], and delaying care and management can negatively impact its clinical evolution. It has been reported that the development of in-hospital (HA) AKI confers a greater risk of mortality than in patients with community-acquired (CA) AKI [8–10], but little is known regarding the outcomes of AKI according to the site of acquisition in COVID patients.

The primary aim of this article is to describe the incidence, risk factors and mortality between CA-AKI and HA-AKI in two reference centers for patients with COVID-19 in Mexico and to develop a simple score that predicts which COVID-AKI patients have a higher risk of dying during hospitalization. Our hypothesis is that CA-AKI carries the same risk of dying as HA-AKI due to the conditions involved in the care of those affected by this pandemic in our country.

## Materials and methods

### Study design and cohort

This was a prospective observational study conducted at the Civil Hospital of Guadalajara Fray Antonio Alcalde, Guadalajara, and the Medical Center 20 de November in Mexico City between April 2020 and February 2021. All adult patients (age ≥ 18 years) who tested positive by polymerase chain reaction of a nasopharyngeal sample for COVID-19 and were hospitalized were eligible. All hospitalized patients had COVID classified as moderate and severe. For patients who had multiple qualifying hospital admissions, we included only the first hospitalization. Patients who were transferred between hospitals within the health system were treated as 1 hospital encounter. Patients were excluded if they were transferred to hospitals out of the health system for which we were unable to obtain data, if they had chronic kidney disease (CKD) grade 5 on dialysis, or if there was a lack of data. Patients who had preexisting CKD G1-G5 not on dialysis and who did not have creatinine measurements during the previous 3 months were also excluded from the analysis. This study was approved by the Institutional Review Board of both hospitals (Hospital Civil de Guadalajara HCG/CEI-0473/20 and Centro

Medico Nacional 20 de Noviembre 09–240.2020) was conducted in adherence to the Declaration of Helsinki. Written informed consent was obtained from all of the subjects at hospitalization. The protocol followed the Strengthening the Reporting of Observational Studies in Epidemiology (STROBE) guidelines [11].

## Data collection, definitions, and measurements

Clinical characteristics, demographic information, and laboratory data were collected prospectively using automated retrieval from the institutional electronic medical record system. COVID severity was categorized as follows: moderate illness, individuals who showed evidence of lower respiratory disease during clinical assessment or imaging and who had saturation of oxygen (SpO2) ≥94% on room air at sea level; and severe illness, individuals who had SpO2 <94% on room air at sea level, a ratio of arterial partial pressure of oxygen to fraction of inspired oxygen (PaO2/FiO2) <300 mm Hg, respiratory frequency >30 breaths/min, or lung infiltrates >50% [12]. AKI was diagnosed by the serum creatinine (Scr) KDIGO criteria [13], and patients were stratified according to the highest AKI stage attained during their hospital stay.

We identified community-acquired AKI when the patient met any of the following criteria: (a) an increased Scr level at admission and a trend of decreasing Scr levels during the hospital stay; (b) an increased Scr level at admission and an Scr level that continued to increase or remained at a high level during the hospital stay, with preadmission Scr values establishing the existence of AKI; or (c) normal kidney function upon admission with Scr levels that began to increase and AKI that could be defined within 2 days after hospitalization combined with causal factors that were determined (by the nephrologists among the investigators) to be present prior to admission based on review of their medical records. If otherwise, AKI was classified as hospital-acquired [14].

The baseline Scr level was defined as the lowest Scr value that was available in the last 3 months prior to admission and throughout the hospital stay. For patients who had no reliable Scr record before admission and no evidence of baseline CKD, a back-estimation of the baseline Scr level was performed based on the 4-variable MDRD (Modification of Diet in Renal Disease) Study equation with the assumption of an estimated glomerular filtration rate of 75 mL/min/1.73 $m^2$ following the recommendations of the 2012 KDIGO AKI clinical practice guideline [14, 15]. The estimated glomerular filtration rate was calculated using the Chronic Kidney Disease Epidemiology Collaboration creatinine equation [16].

We collected data on patient demographics, baseline history of comorbid conditions, and home medications. Comorbid conditions were determined from provider-entered past medical history and admission medication reconciliation. In addition to baseline clinical data, we collected information from the time of hospital admission, such as hemoglobin, platelets, leukocytes, glucose, urea, creatinine, sodium, potassium, chloride, phosphate, calcium, arterial pH, PCO2, PO2, bicarbonate and lactate levels. We also recorded the need for invasive mechanical ventilation, length of stay, and the need for KRT. Indications for KRT were fluid overload resistant to diuretics, severe hyperkalemia, severe metabolic acidosis, and uremic manifestations, including encephalopathy, pericarditis, and convulsions [17, 18].

## Outcomes

The primary outcome was 28 days survival compared between hospitalized COVID patients with CA-AKI and HA-AKI. Secondary outcomes included the need for KRT, risk factors associated with the development of CA-AKI and HA-AKI, and the creation of a predictive mortality score for hospitalized COVID-AKI patients.

## Statistical analysis

Continuous variables are summarized as the mean ± SD unless otherwise specified. Categorical variables are summarized as numbers with percentages. We compared baseline patient characteristics between patients with or without AKI and with CA-AKI or HA-AKI using Fisher exact tests for categorical variables and nonparametric Wilcoxon signed-rank tests for continuous variables. Survival after hospital admission was estimated using Kaplan–Meier plots and compared using the log-rank test. To identify risk factors associated with mortality, KRT, CA-AKI, and HA-AKI, we constructed a logistic regression model adjusted for age, sex, COVID severity, diabetes, hypertension, CKD, stroke, COPD, BMI, ferritin, leukocytes, lymphocytes, mechanical ventilation, KRT, AKI stage, and AKI acquisition. Only the independent variables found to have $P < 0.05$ based on univariate analysis were included in the multivariate analysis. $P < 0.05$ was considered significant. A predictive mortality score was developed, which we called SARS-AKI. Variables that were statistically significant in the multivariate logistic regression method were included in the SARS-AKI score. To quantify the goodness of fit of our prediction model, we used the area under the receiver operating characteristic (ROC) curve and calibration. The statistical analyses were performed using R (version 3.6.3; R Foundation for Statistical Computing, Vienna, Austria).

## Results

From April 1, 2020, to February 1, 2021, 1251 patients were admitted to the Civil Hospital of Guadalajara Fray Antonio Alcalde, Guadalajara, and the Medical Center 20 de Noviembre in Mexico City with a diagnosis of COVID-19 present on admission or made during hospitalization. Of these, 877 were used as the analysis cohort (Fig 1). The baseline characteristics of the study cohort by AKI and no AKI are provided in Table 1. A total of 377 patients (33.7%) developed AKI. Patients with AKI were significantly older, predominantly male, had more comorbidities, increased markers of inflammatory response, more severe COVID-19, an increased need for mechanical ventilation, and increased mortality.

Among the patients with AKI, we observed that CA-AKI occurred in 202 patients (59.9%) and HA-AKI occurred in 135 (40.1%). Patients with CA-AKI had more significant comorbidities, including diabetes (52.4% vs 38.5%), hypertension (58.4% vs 39.2%), CKD (30.1% vs 14.8%), and COPD (5.9% vs 1.4%), than those with HA-AKI. CA-AKI also presented with less significant lymphocytes, hemoglobin, and bicarbonate levels, a greater increase in serum potassium, and a greater need for KRT. No significant difference was observed in AKI stages, the need for mechanical ventilation, or mortality (Table 2).

The overall 28-day survival among hospitalized COVID patients was 68.0% (IC 95% 0.65–0.71) (S1 Fig). The 28-day survival comparing CA-AKI, HA-AKI, and mechanical ventilation is presented in Fig 2. Patients' survival without AKI was 87.1% (IC 95% 0.84–0.90), with CA-AKI it was 75.4% (IC 95% 0.68–0.83), with HA-AKI it was 69.6% (IC 95% 0.59–0.81), and with mechanical ventilation it was 30.1% (IC 95% 0.21–0.42). Patient survival with CA-AKI and mechanical ventilation was 21.2% (IC 95% 0.13–0.32), while with HA-AKI and mechanical ventilation it was 16.7% (IC 95% 0.09–0.28), log-rank test p < 0.001. When comparing the 28-day survival between AKI stages and mechanical ventilation, patients' survival with AKI stages 2–3 was 68.5% (IC 95% 0.60–0.77), and with mechanical ventilation plus AKI stages 2–3 it was 17.2% (IC 95% 0.11–0.25) (S2 Fig).

The mortality percentage according to AKI stage, CA-AKI, and HA-AKI is presented in Fig 3. Patients with stage 3 AKI and CA-AKI had a mortality of 48.9%, while patients with stage 3 AKI and HA-AKI had a mortality of 52.6%, p = 0.12.

**Assessed for eligibility:**

1. Hospital admissions with **positive COVID-19** tests between April 2020 and February 2021 (data from 2 hospitals)

2. Patients included: age ≥ 18 years

N = 1251

**374 patients excluded:**

1. Chronic kidney disease grade 5 on dialysis
   N = 20 (1.5%)
2. Chronic kidney disease G1 to G5 not on dialysis without Scr for the previous 3 months
   N = 4 (0.3%)
3. Multiple admissions for 1 patient; only the first episode of admission was included
   N= 15 (1.1%)
4. Hospital transfers out of our health system
   N= 105 (8.3%)
5. Lack of data
   N= 230 (18.3%)

**Patients accepted in the analysis**

N = 877

**Fig 1. Flowchart of study population.**

**Table 1. Baseline characteristics of the study cohort by AKI and no AKI.**

| Variable | Total (N = 877) | No AKI (N = 540) | AKI (N = 337) | p value |
|---|---|---|---|---|
| Age [years], mean (SD) | 55.9 (15.7) | 54.5 (15.6) | 58.2 (15.6) | <0.001* |
| Male [N (%)] | 548 (62.5) | 321 (59.4) | 227 (67.3) | 0.02* |
| Body Mass Index [kg/m$^2$], mean (SD) | 28.1 (6.3) | 28.3 (6.2) | 27.7 (6.5) | 0.06 |
| Comorbidities [N (%)] | | | | |
| Diabetes | 311 (35.4) | 153 (28.3) | 158 (46.8) | <0.001* |
| Hypertension | 332 (37.8) | 161 (29.8) | 171 (50.7) | <0.001* |
| CKD | 97 (11) | 16 (2.9) | 81 (24) | <0.001* |
| Stroke | 37 (4.2) | 15 (2.7) | 22 (6.5) | 0.01* |
| COPD | 27 (3.0) | 13 (2.4) | 14 (4.1) | 0.14 |
| Admission biochemical data, mean (SD) | | | | |
| Serum Ferritin [ng/m] | 1144 (2068) | 940 (2350) | 1484 (2962) | <0.001* |
| Serum Leukocytes [$10^9$/L] | 4.2 (5.2) | 4.1 (4.7) | 4.4 (6.0) | 0.98 |
| Serum Lymphocytes [μL] | 1130 (1297) | 1190 (784) | 1037 (1824) | <0.001* |
| Serum Hemoglobin [g/dL] | 13.7 (2.3) | 14.1 (2.1) | 13.2 (2.6) | <0.001* |
| Serum Platelets [$10^9$/L] | 245 (156) | 249 (100) | 240 (216) | 0.56 |
| Serum Sodium [mEq/L] | 136 (7.4) | 136.3 (8.1) | 135.5 (6.2) | 0.26 |
| Serum Potassium [mEq/L] | 4.1 (0.8) | 4 (0.7) | 4.4 (0.9) | <0.001* |
| Arterial pH | 7.41 (0.1) | 7.43 (0.1) | 7.40 (0.1) | 0.76 |
| Serum bicarbonate [mEq/L] | 21.6 (4.6) | 22.2 (4.5) | 20.8 (4.6) | 0.001* |
| Serum Creatinine [mg/dL] | 1.7 (3.3) | 0.8 (0.4) | 2.9 (5.0) | <0.001* |
| COVID severity [N (%)] | | | | <0.001* |
| Mild | 511 (58.2) | 370 (68.5) | 141 (41.8) | |
| Severe | 366 (41.8) | 170 (31.5) | 196 (58.2) | |
| Need in mechanical lung ventilation [N (%)] | 219 (24.9) | 73 (13.5) | 146 (43.3) | <0.001* |
| Length of stay, days, mean (SD) | 9.9 (6.7) | 9.0 (6.2) | 11.4 (7.2) | <0.001* |
| Disposition [N (%)] | | | | <0.001* |
| Discharged | 597 (68.1) | 429 (79.4) | 168 (49.8) | |
| Expired | 280 (31.9) | 111 (20.6) | 169 (50.2) | |

AKI, acute kidney injury; CKD, chronic kidney disease; COPD, chronic obstructive pulmonary disease; KRT, kidney replacement therapy.

In an attempt to identify variables associated with mortality at the 28-day follow-up, univariate and multivariate analyses were performed. Only age > 60 years (OR 1.12, 95% CI 1.06–1.18, p <0.001), COVID severity (OR 1.09, 95% CI 1.03–1.16, p = 0.002), the need for mechanical ventilation (OR 1.67, 95% CI 1.56–1.78, p <0.001), and HA-AKI stage 3 (OR 1.16, 95% CI 1.05–1.29, p = 0.003) had a significant increase in mortality (Table 3). The presence of CKD (OR 1.48, 95% CI 1.391.56, p < 0.001), serum lymphocytes < 1000 μL (OR 1.03, 95% CI 1.00–1.07, p = 0.03), the need for mechanical ventilation (OR 1.06, 95% CI 1.02–1.11, p = 0.003), and CA-AKI stage 3 (OR 1.37, 95% CI 1.29–1.46, p < 0.001) were the only variables associated with a KRT start in the multivariable analysis (Table 3).

On the other hand, it was observed that having severe COVID (OR 1.08, 95% CI 1.02–1.14, p = 0.005), diabetes (OR 1.08, 95% CI 1.02–1.15, p = 0.005), hypertension (OR 1.07, 95% CI 1.00–1.14, p = 0.02), CKD (OR 1.47, 95% CI 1.35–1.60, p < 0.001), COPD (1.25, 95% CI 1.07–1.45, p = 0.003), and serum leucocytes > 12 109/L (OR 1.10, 95% CI 1.04–1.17, p < 0.001) were associated with the development of CA-AKI, while only the presence of severe COVID (OR 1.13, 95% CI 1.06–1.20, p < 0.001), and serum ferritin > 500 [ng/m] (OR 1.07, 95% CI 1.01–1.14, p = 0.01) were significant factors for the development of HA-AKI (Table 4).

**Table 2. Baseline characteristics by AKI acquisition.**

| Variable | AKI acquisition | | p-value |
|---|---|---|---|
| | CA-AKI (N = 202) | HA-AKI (N = 135) | |
| Age [years], mean (SD) | 58.1 (15.7) | 58.4 (15.4) | 0.74 |
| Male [N (%)] | 132 (65.3) | 95 (70.3) | 0.33 |
| Body Mass Index [kg/m$^2$], mean (SD) | 27.7 (7.6) | 27.5 (4.5) | 0.23 |
| Comorbidities [N (%)] | | | |
| Diabetes | 106 (52.4) | 52 (38.5) | 0.01* |
| Hypertension | 118 (58.4) | 53 (39.2) | <0.001* |
| CKD | 61 (30.1) | 20 (14.8) | 0.001* |
| Stroke | 13 (6.4) | 9 (6.6) | 0.64 |
| COPD | 12 (5.9) | 2 (1.4) | 0.04* |
| Admission biochemical data, mean (SD) | | | |
| Serum Ferritin [ng/m] | 1325 (1259) | 1725 (4424) | 0.47 |
| Serum Leukocytes [$10^9$/L] | 4.0 (6.3) | 5.0 (5.5) | 0.09 |
| Serum Lymphocytes [µL] | 1032 (2139) | 1044 (1122) | 0.04* |
| Serum Hemoglobin [g/dL] | 12.8 (2.9) | 13.8 (1.8) | 0.003* |
| Serum Platelets [$10^9$/L] | 231 (101) | 255 (328) | 0.70 |
| Serum Sodium [mEq/L] | 135 (6.6) | 136.3 (5.3) | 0.46 |
| Serum Potassium [mEq/L] | 4.6 (1.0) | 4.1 (0.6) | <0.001* |
| Arterial pH | 7.37 (0.1) | 7.41 (0.1) | 0.07 |
| Serum bicarbonate [mEq/L] | 19.9 (4.5) | 21.9 (4.5) | 0.01* |
| Serum Creatinine [mg/dL] | 3.8 (5.9) | 1.3 (1.7) | <0.001* |
| COVID severity [N (%)] | | | 0.43 |
| Mild | 88 (43.5) | 53 (39.2) | |
| Severe | 114 (56.5) | 82 (60.8) | |
| AKI | | | |
| KDIGO-1 | 64 (31.6) | 41 (30.3) | 0.87 |
| KDIGO-2 | 49 (24.2) | 33 (24.4) | 0.88 |
| KDIGO-3 | 89 (44.1) | 61 (45.1) | 0.76 |
| KRT | 60 (29.7) | 27 (20.0) | 0.04* |
| Need in mechanical lung ventilation [N (%)] | 80 (39.6) | 66 (48.8) | 0.09 |
| Length of stay, days, mean (SD) | 11.0 (7.4) | 12.0 (7.4) | 0.20 |
| Disposition [N (%)] | | | 0.06 |
| Discharged | 109 (53.9) | 59 (43.7) | |
| Expired | 93 (46.1) | 76 (56.3) | |

AKI, acute kidney injury; CA-AKI, community-acquired AKI; HA-AKI, hospital-acquired AKI; CKD, chronic kidney disease; COPD, chronic obstructive pulmonary disease; KRT, kidney replacement therapy.

We developed the predictive SARS-AKI score as a tool for predicting mortality at 28 days in COVID patients. We included age > 60 years, severe COVID, the need for mechanical ventilation, AKI stage, and AKI acquisition (Fig 4). A cutoff value ≥ 3 points had an AUC of 0.82 (95% CI 1.15–1.18), p < 0.001, with a specificity of 0.87, sensitivity of 0.66, PPV of 0.71, and an NPV of 0.84.

## Discussion

In this multicenter prospective cohort, we found that COVID patients who are complicated by CA-AKI have more comorbidities and worse biochemical parameters at the time of

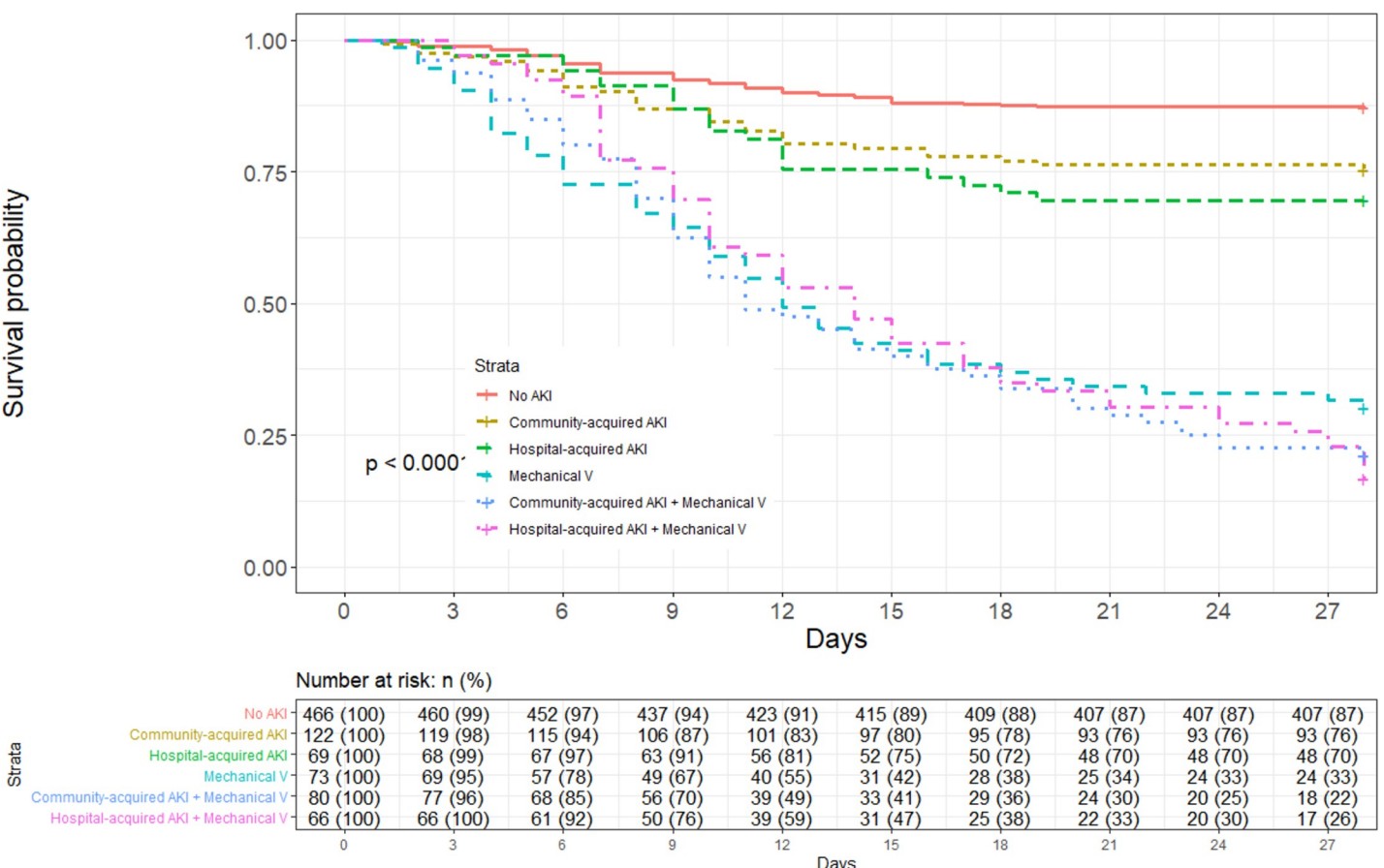

**Fig 2. Kaplan–Meier survival statistics comparing community-acquired AKI, hospital-acquired AKI, and mechanical ventilation.** Numbers of patients at risk at each time point shown below the graph.

hospitalization than HA-AKI patients, but despite these differences, their probability of dying is similar. In addition, when these patients needed MV, the prognosis in both groups worsened markedly, and we were able to develop a predictive score for death among patients with AKI COVID.

Notably, the mortality of patients with COVID and AKI was not different between the HA-AKI and CA-AKI subgroups. Even when comparing the CA-AKI and HA-AKI KDIGO stages, there were no differences in mortality between them, results that could be contrasted with previous studies. In our cohort, the mortality of patients with AKI at 28 days was 68%, higher than that reported by the STOP-COVID Investigators group, where they described that 54.9% of AKI COVID patients in the ICU died within 28 days of ICU admission [19], and more likely to the 60.1% mortality rate reported in a Sau Paulo cohort [20]. These results can be explained by the conditions, hospital infrastructure and health system of our country compared to the US and Brazil.

Studying the distinction in mortality between CA-AKI and HA-AKI is relevant in the context of COVID because AKI due to other causes has historically shown epidemiological, clinical and prognostic differences between these two groups [8–10]. It is possible that the natural trajectory of COVID-19 with fever, nausea, vomiting, fatigue, anorexia, self-medication, fear of imminent hospitalization and lack of hospital beds for their care has resulted in patients in

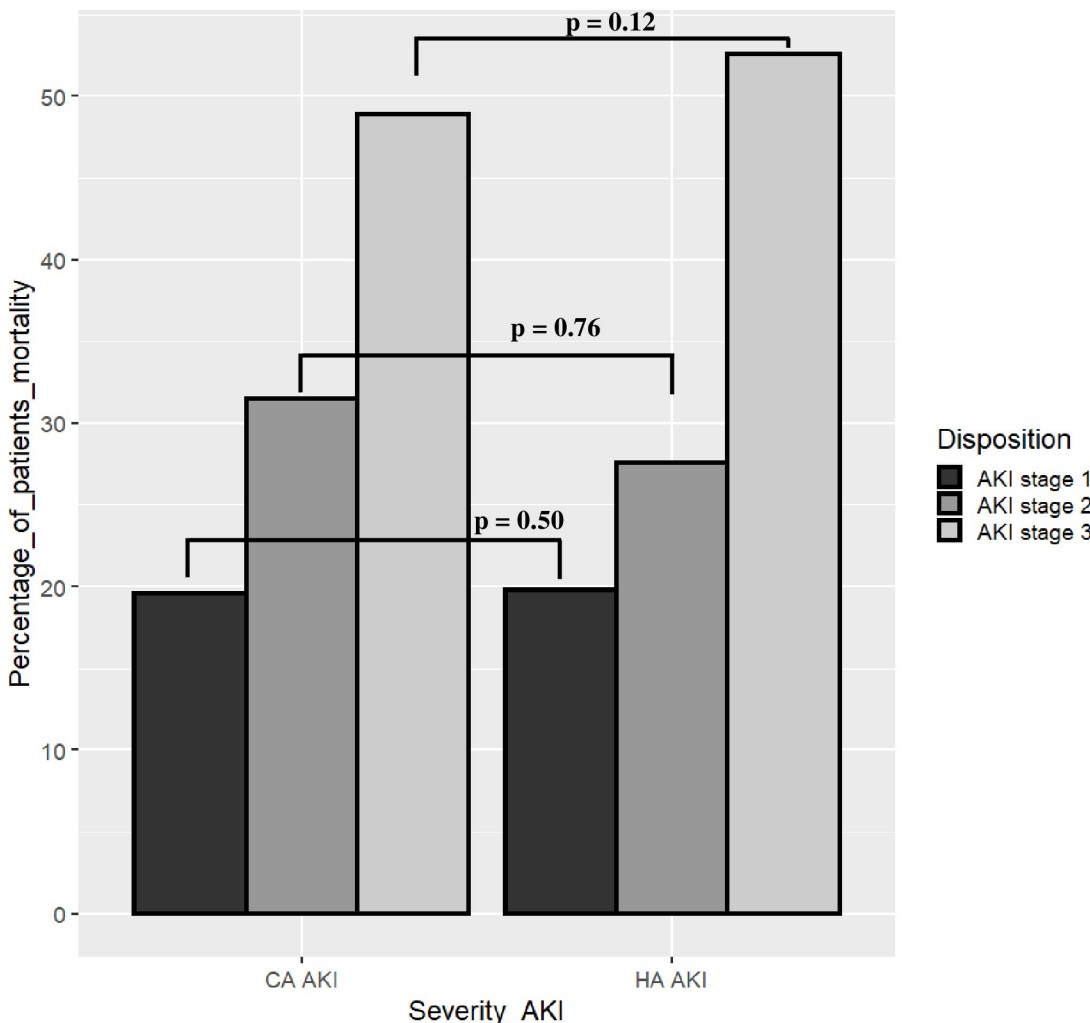

**Fig 3. Mortality percent according to AKI stage, CA-AKI, and HA-AKI.**

our cohort arriving late to the hospital, and this time lost by delaying its management also helps explain these unfavorable results [21].

In our cohort, CA-AKI patients more frequently had hypertension, CKD and COPD than HA-AKI patients, antecedents that have been linked to a worse clinical evolution in patients with COVID [22, 23], including the development of AKI [4].

The body mass index (BMI) in our cohort was 28.1 (overweight), we found no association between BMI and the development of COVID AKI or its complications, in contrast to previous meta-analysis where 9 cohorts were included and two of them considered BMI> 24, and reported that obesity increases the severity of COVID [24], this lack of association could be explained by the fact that most of the evidence has reported obesity, not overweight, as a risk factor for complications from COVID. The comorbidities that we found in patients with CA-AKI are very similar to those described in the Mexican cohort by Martínez-Rueda et al. [25], where it was found that patients with COVID CA-AKI have older age and a greater number of comorbidities compared to patients with HA-AKI; results that validate ours findings since they are patients from the same region.

**Table 3. Factors associated with mortality at 28-days of follow up and the requirement of kidney replacement therapy in hospitalized COVID patients, in the univariable and multivariable logistic regression model.**

| | Univariable analysis, OR (95% CI) | p | Multivariable analysis, OR (95% CI) | p |
|---|---|---|---|---|
| **Mortality** | | | | |
| Age > 60 [years] | 1.20 (1.13–1.27) | <0.001 | 1.12 (1.06–1.18) | <0.001 |
| COVID severity | | | | |
| Mild | Reference | - | Reference | - |
| Severe | 1.41 (1.33–1.49) | <0.001 | 1.09 (1.03–1.16) | 0.002 |
| Need in mechanical lung ventilation | 1.82 (1.72–1.93) | <0.001 | 1.67 (1.56–1.78) | <0.001 |
| AKI acquisition and KDIGO stage | | | | |
| CA-AKI and stage 2–3 | Reference | - | Reference | - |
| HA-AKI and stage 2 | 1.38 (1.18–1.63) | <0.001 | 1.11 (0.97–1.27) | 0.10 |
| HA-AKI and stage 3 | 1.43 (1.27–1.61) | <0.001 | 1.16 (1.05–1.29) | 0.003 |
| **Kidney replacement therapy** | | | | |
| CKD | 1.69 (1.59–1.78) | <0.001 | 1.48 (1.39.1.56) | <0.001 |
| Serum Lymphocytes < 1000 [μL] | 1.10 (1.06–1.15) | <0.001 | 1.03 (1.00–1.07) | 0.03 |
| Need in mechanical lung ventilation | 1.12 (1.07–1.18) | <0.001 | 1.06 (1.02–1.11) | 0.003 |
| AKI acquisition and KDIGO stage | | | | |
| HA-AKI and stage 2–3 | Reference | - | Reference | - |
| CA-AKI and stage 2 | 0.99 (0.90–1.08) | 0.90 | | |
| CA-AKI and stage 3 | 1.66 (1.56–1.76) | <0.001 | 1.37 (1.29–1.46) | <0.001 |

AKI, acute kidney injury; CA-AKI, community-acquired AKI; HA-AKI, hospital-acquired AKI; CKD, chronic kidney disease; COPD, chronic obstructive pulmonary disease; KRT, kidney replacement therapy.

Which could explain why these patients, with a susceptible kidney, quickly develop AKI in the face of COVID. Pelayo et al. highlighted that patients with CA-AKI have fewer complications during hospitalization than those with HA-AKI, such as the need for vasopressors, MV and death (23% vs 52%) [26]. These data differ from our cohort, since in our study, there was

**Table 4. Factors associated with CA-AKI and HA-AKI in hospitalized COVID patients, in the univariable and multivariable logistic regression model.**

| | Univariable analysis, OR (95% CI) | p | Multivariable analysis, OR (95% CI) | p |
|---|---|---|---|---|
| **Community-acquired AKI** | | | | |
| Grade COVID | | | | |
| Mild | Reference | - | Reference | - |
| Severe | 1.15 (1.08–1.21) | <0.001 | 1.08 (1.02–1.14) | 0.005 |
| Diabetes | 1.18 (1.12–1.25) | <0.001 | 1.08 (1.02–1.15) | 0.005 |
| Hypertension | 1.22 (1.15–1.29) | <0.001 | 1.07 (1.00–1.14 | 0.02 |
| CKD | 1.56 (1.43–1.70) | <0.001 | 1.47 (1.35–1.60) | <0.001 |
| COPD | 1.24 (1.06–1.46) | 0.007 | 1.25 (1.07–1.45) | 0.003 |
| Serum Leukocytes > 12 [$10^9$/L] | 1.14 (1.07–1.21) | <0.001 | 1.10 (1.04–1.17) | <0.001 |
| **Hospital-acquired AKI** | | | | |
| COVID severity | | | | |
| Mild | Reference | - | Reference | - |
| Severe | 1.12 (1.07–1.18) | <0.001 | 1.13 (1.06–1.20) | <0.001 |
| Serum Ferritin > 500[ng/m] | 1.10 (1.04–1.16) | <0.001 | 1.07 (1.01–1.14) | 0.01 |

AKI, acute kidney injury; CA-AKI, community-acquired AKI; HA-AKI, hospital-acquired AKI; CKD, chronic kidney disease; COPD, chronic obstructive pulmonary disease.

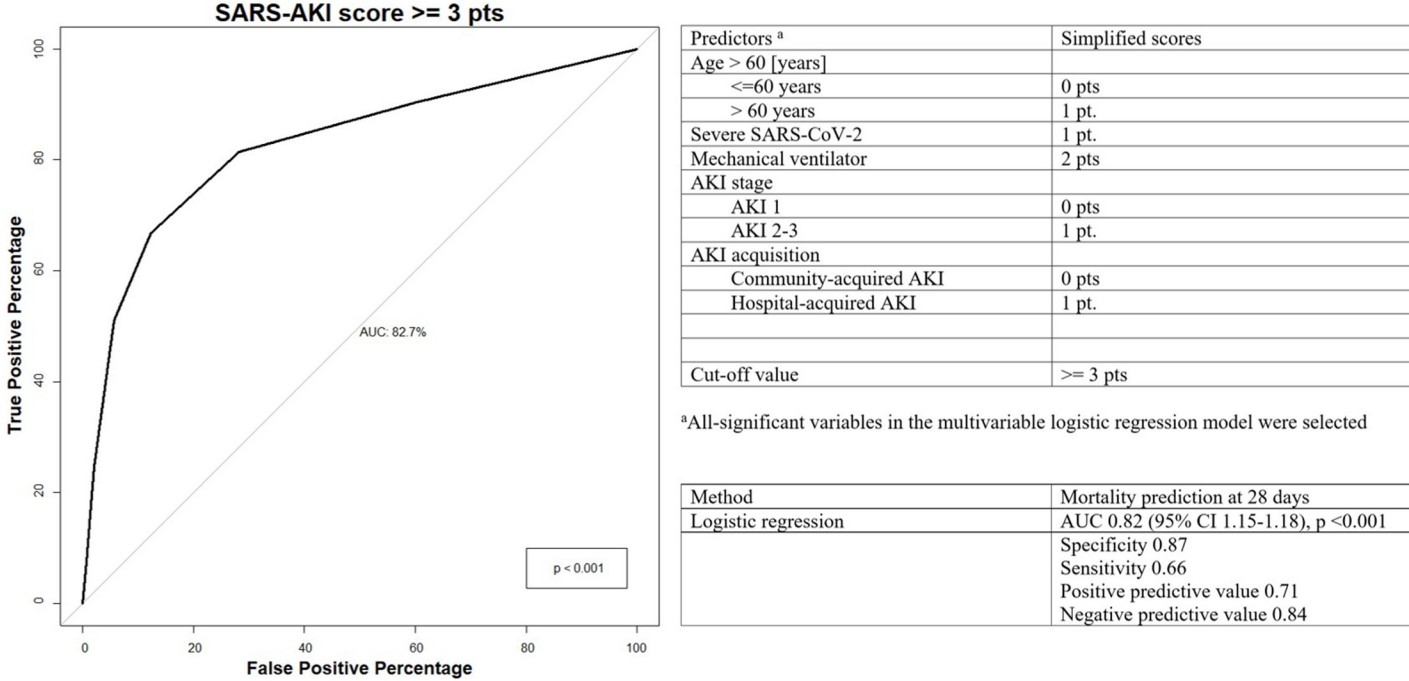

| Predictors [a] | Simplified scores |
|---|---|
| Age > 60 [years] | |
| <=60 years | 0 pts |
| > 60 years | 1 pt. |
| Severe SARS-CoV-2 | 1 pt. |
| Mechanical ventilator | 2 pts |
| AKI stage | |
| AKI 1 | 0 pts |
| AKI 2-3 | 1 pt. |
| AKI acquisition | |
| Community-acquired AKI | 0 pts |
| Hospital-acquired AKI | 1 pt. |
| | |
| Cut-off value | >= 3 pts |

[a]All-significant variables in the multivariable logistic regression model were selected

| Method | Mortality prediction at 28 days |
|---|---|
| Logistic regression | AUC 0.82 (95% CI 1.15-1.18), p <0.001 |
| | Specificity 0.87 |
| | Sensitivity 0.66 |
| | Positive predictive value 0.71 |
| | Negative predictive value 0.84 |

**Fig 4. COVID-AKI score as a tool for predicting death in COVID AKI patients and the ROC curve of the logistic regression model for predicting all-cause mortality.**

no difference in the need for MV between CA-AKI and HA-AKI. It is striking that in the subgroup with CA-AKI, more patients were intubated (39.6%), and it is likely that this complication explains that in our cohort, twice as many patients (46%) died compared to only 23% in the Pelayo et al. study [26]. We also described the impact that the combination of AKI and VM has on mortality. We found a poor prognosis in this scenario with a mortality rate of only 17.2%.

The adverse renal effects of mechanical ventilation have been studied [27, 28]. The risk of developing AKI in mechanically ventilated patients has been demonstrated in various entities: in cardiovascular disease [29]; gastrointestinal disease [30]; even in past pandemics like H1N1 [31]. Where the exposed to varying levels of continuous positive airway pressure causes a reduction in cardiac output by impeding venous return, circulatory stress, and neurohumoral mediators released that alter renal blood flow from cortex to medulla which leads to sodium reabsorption and a reduction in the glomerular filtration rate [32]. More recently, mechanical ventilation induced lung injury (VILI) has been proposed as another mechanism of AKI via inflammatory crosstalk from the lung to kidney. Although mechanical ventilation has a major role in life support, evidence suggests that certain ventilator settings may induce lung injury in some cases or may worsen lung injury once established [33]. Our study provides data on epidemiological aspects as well as the etiology and outcomes of COVID-AKI and adds to the literature the important differences between CA-AKI and HA-AKI.

We identified risk factors for the development of CA-AKI, is likely to be multifactorial, with cardiovascular comorbidity and predisposing factors as important contributors [27] including comorbidities such as diabetes, hypertension, CKD and COPD, and debilitating chronic diseases that have been universally associated with a worse evolution of COVID 19. In addition, we also identified that the severity of COVID and markers linked to inflammation, which is consistent with the pathophysiological mechanisms described in its pathology, where

high inflammatory activity plays a preponderant role in kidney injury [34], evident in high levels of leukocytes (> 12 109/L) and ferritin (> 500 ng/mL), similar to ferritin levels (798 ng/mL) previously described by the Northwell COVID-19 Research Consortium [22] and similar leukocytes than the Brazilian cohort [20] cohort.

Patients with CA-AKI needed KRT more frequently than patients with HA-AKI (29.7% vs 20.0%), with exactly the same frequency as that reported in the EPILAT-IRA study, where KRT was performed in 29% of cases. Relevant data in this context were obtained because, as in our population, this is a Latin-American study with a high prevalence of CA AKI 63% [35]. We identified that having CKD increased the need for KRT by almost 50%, a result that is not surprising since CKD is considered the greatest risk factor for AKI and its complications [36]. Additionally, receiving MV increased the risk of KRT; again, a widely described event, the kidney and lung are closely related, and having MV increases the risk of AKI 3-fold. High levels of inflammation triggered by lung damage could affect kidney function [37].

The high frequency of KRT in the CA-AKI group can be justified by the presence of more comorbidities (hypertension, CKD and COPD) and markers of inflammation such as lymphopenia, anemia and electrolyte disturbance related to AKI-specific complications such as hyperkalemia, hypobicarbonatemia, and azotemia, all of which may have been triggers for the initiation of KRT.

The ADQI group recommends identifying COVID patients who are at high risk of developing AKI and its complications [2]. To our knowledge, a predictive death score has never been described in patients with AKI COVID. In this study, we developed a risk score derived from patient demographics, COVID severity and acute risk factors and demonstrated that it can reliably predict mortality in AKI COVID patients, with an AUC of 0.82 with the optimal cutoff for mortality was estimated to be $\geq$ 3 points (Fig 4). Implementation of this risk model in clinical practice may help target high-risk patients for surveillance and enable clinicians to evaluate novel diagnostic, preventive and therapeutic modalities to mitigate mortality in AKI COVID.

Our study has limitations that must be interpreted together with its results, although its retrospective design may have failed to capture relevant variables, such as the treatment offered. The incidence of AKI might be underestimated due to the real-world nature of data collection. As expected, there was uncertainty regarding the time of onset of AKI before admission among those with CA-AKI. Mortality was only considered during the 28 days of hospitalization and a longer follow-up would have been appropriate since many patients have prolonged hospital stays.

Finally, the possibility of residual confounding exists for our observational study.

Our study has the strength of a relatively large sample size and detailed information on various variables obtained from two of the largest tertiary hospitals in Mexico. We divided the analysis between patients with HA-AKI and CA-AKI, a topic little explored until now.

In conclusion, in this multicenter prospective cohort, we found that patients with COVID who are complicated with CA-AKI have more comorbidities, and despite this, they have the same evolution when compared with HA AKI. We also developed a death predictive score for patients with AKI COVID.

## Supporting information

**S1 Fig. Overall, 28-days survival in hospitalized COVID patients.**
(DOCX)

**S2 Fig. Kaplan–Meier survival statistics comparing AKI stages and mechanical ventilation.**
Numbers of patients at risk at each time point shown below the graph.
(DOCX)

**S1 Data.**
(XLSX)

# Acknowledgments

The authors would like to acknowledge the medical, nursing, and allied health staff of both hospitals for treating all patients with joy and kindness.

# Author Contributions

**Conceptualization:** Jonathan S. Chávez-Íñiguez, José H. Cano-Cervantes, Pablo Maggiani-Aguilera, Sandra Muñoz-López, Olivia Montoya-Montoya, Guillermo Navarro-Blackaller, Aczel Sánchez-Cedillo.

**Data curation:** Jonathan S. Chávez-Íñiguez, José H. Cano-Cervantes, Pablo Maggiani-Aguilera, Natashia Lavelle-Góngora, Cynthia Moreno-González, Jarumi A. Tanaka-Gutiérrez, Ana P. Villa Zaragoza, Karla E. Rincón-Souza, Sandra Muñoz-López, Olivia Montoya-Montoya, Aczel Sánchez-Cedillo.

**Formal analysis:** Jonathan S. Chávez-Íñiguez, José H. Cano-Cervantes, Pablo Maggiani-Aguilera.

**Investigation:** Jonathan S. Chávez-Íñiguez, José H. Cano-Cervantes, Pablo Maggiani-Aguilera, Natashia Lavelle-Góngora, Josué Marcial-Meza.

**Methodology:** Jonathan S. Chávez-Íñiguez, José H. Cano-Cervantes, Pablo Maggiani-Aguilera.

**Project administration:** Jonathan S. Chávez-Íñiguez, Pablo Maggiani-Aguilera.

**Supervision:** Jonathan S. Chávez-Íñiguez, José H. Cano-Cervantes, Pablo Maggiani-Aguilera, Josué Marcial-Meza, Guillermo García-García.

**Validation:** Jonathan S. Chávez-Íñiguez, José H. Cano-Cervantes, Pablo Maggiani-Aguilera, Josué Marcial-Meza, Luis E. Morales-Buenrostro, Guillermo García-García.

**Visualization:** Jonathan S. Chávez-Íñiguez, José H. Cano-Cervantes, Pablo Maggiani-Aguilera, Natashia Lavelle-Góngora, Estefanía P. Camacho-Murillo, Luis E. Morales-Buenrostro, Guillermo García-García.

**Writing – original draft:** Jonathan S. Chávez-Íñiguez, José H. Cano-Cervantes, Pablo Maggiani-Aguilera.

**Writing – review & editing:** Jonathan S. Chávez-Íñiguez, José H. Cano-Cervantes, Pablo Maggiani-Aguilera.

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
