## [Decision Letter · Decision Letter 0]

11 Jun 2021

PONE-D-21-11844

Mortality and evolution between Community and Hospital-acquired COVID-AKI

PLOS ONE

Dear Dr. Chávez-Iñiguez,

Thank you for submitting your manuscript to PLOS ONE. After careful consideration, we feel that it has merit but does not fully meet PLOS ONE’s publication criteria as it currently stands. Therefore, we invite you to submit a revised version of the manuscript that addresses the points raised during the review process.

As the Academic Editor, I have sent your manuscript for revision to several persons. Currently one expert in the field has already provided the peer-review with a very detailed and useful comments, suggesting Major Revision. Other experts invited so far have declined the invitation to review the manuscript (it is rather widespread situation, related to different factors).

I would like to save your time, and provide the possibility to perform the revision of the manuscript based on the feedback from one expert in the field, and make it stronger. Once you submit the revised manuscript, I will send it to the expert who provided the feedback, and also involve another reviewer. If you have any objections for this approach, please let me know. If you agree, please proceed this way and submit the revised manuscript along with the point-by-point responses to the reviewer. 

We look forward to receiving your revised manuscript.

Kind regards,

Boris Bikbov

Academic Editor

PLOS ONE

Journal Requirements:

2.Please provide additional details regarding participant consent. In the ethics statement in the Methods and online submission information, please ensure that you have specified what type you obtained (for instance, written or verbal, and if verbal, how it was documented and witnessed). If your study included minors, state whether you obtained consent from parents or guardians. If the need for consent was waived by the ethics committee, please include this information.

3. Thank you for providing the date(s) when patient medical information was initially recorded. Please also include the date(s) on which your research team accessed the databases/records to obtain the retrospective data used in your study

4.Upon re-submitting your revised manuscript, please upload your study’s minimal underlying data set as either Supporting Information files or to a stable, public repository and include the relevant URLs, DOIs, or accession numbers within your revised cover letter. For a list of acceptable repositories, please see http://journals.plos.org/plosone/s/data-availability#loc-recommended-repositories. Any potentially identifying patient information must be fully anonymized.

Additional Editor Comments (if provided):

Reviewers' comments:

Reviewer's Responses to Questions

**Comments to the Author**

1. Is the manuscript technically sound, and do the data support the conclusions?

Reviewer #1: Yes

2. Has the statistical analysis been performed appropriately and rigorously? 

Reviewer #1: Yes

3. Have the authors made all data underlying the findings in their manuscript fully available?

Reviewer #1: Yes

4. Is the manuscript presented in an intelligible fashion and written in standard English?

Reviewer #1: Yes

5. Review Comments to the Author

Reviewer #1: This is an interesting retrospective study highlighting the differences between HA AKI and CA AKI in COVID patients. The findings are in contrast to that of literature where HA AKI due to usual causes have been found to have worse outcome due to the number of comorbidities. Interestingly, the difference of > 10% in mortality between CA AKI and HA AKI did not show significance. The authors have 28-day mortality as the primary outcome. Does the table 1 show 28-day outcome as some patients may still be in-patient as we know COVID patients have longer length of stay. Do the authors have in-hospital mortality as FU after discharge may result in loss of follow up. Loss of 18.3% of study population due to loss of creatinine is surprising. Do decide about baseline creatinine, the authors have gone back a year but to get a baseline for patients who had pre-existing CKD G1-G5 not on dialysis, only previous 3 months creatinine was necessary, otherwise they were also excluded from the analysis

The other observation is that patients with first admission were included. This creates a survival bias as patient may have died in subsequent admission

The authors have used variable end-points. On some occasions, the authors report survival and on other occasion, mortality. The literature reports mortality for AKI. For example: The multivariable regression is using mortality as dependent variable which survival is quoted for HA & CA AKI

Was AKI not defined by KDIGO classification? This is not stated in methods

6. PLOS authors have the option to publish the peer review history of their article (what does this mean?). If published, this will include your full peer review and any attached files.

Reviewer #1: **Yes: **Nitin V Kolhe

---

## [Author Response · Author response to Decision Letter 0]

25 Jun 2021

Response to Reviewers'

We appreciate the time and the review process, without a doubt the comments of the editors are all relevant and pertinent, they contribute to making our manuscript stronger. We have done our best to satisfy each of them.

Reviewer #1: 

1.-The findings contrast with that of literature where HA AKI due to usual causes have been found to have worse outcome due to the number of comorbidities. 

R= We are glad that the editor has noted this point, from our perspective it is also a valuable argument in our manuscript, and we give a detailed explanation of why this result happened in page 21, 22.

2.-Interestingly, the difference of > 10% in mortality between CA AKI and HA AKI did not show significance. 

R= We believe that the limited number of the event could have explained the lack of statistical significance (p = 0.06), although the trend is notable.

3.- The authors have 28-day mortality as the primary outcome. Does the table 1 show 28-day outcome as some patients may still be in-patient as we know COVID patients have longer length of stay. Do the authors have in-hospital mortality as FU after discharge may result in loss of follow up. 

R= We understand the point, without a doubt the long-term follow-up would be relevant, but we only took into account the first 28 days of the stay in the hospital, unfortunately we could not follow the patients once they were discharged from the hospital, we faced the reality where COVID patients hardly come for follow-up.

If the reviewer agree, we could add the following sentence to the limitations (page 24):

Mortality was only considered during the 28 days of hospitalization and a longer follow-up would have been appropriate since many patients have prolonged hospital stays.

4.- Loss of 18.3% of study population due to loss of creatinine is surprising. 

R= We excluded 230 (18.3%) patients from the analysis, mainly because despite being hospitalized, they did not have serum creatinine measurement, we know the importance of this variable, but in our hospitals the systematic request of serum creatinine is not performed unless the treating physicians request it. 

It is probable that the overload of work and the saturation of the clinical laboratories during pandemic has contributed as well.

5.-Do decide about baseline creatinine, the authors have gone back a year but to get a baseline for patients who had pre-existing CKD G1-G5 not on dialysis, only previous 3 months creatinine was necessary, otherwise they were also excluded from the analysis

R= We agree with the reviewer's comment, we only need 3 months was necessary, we will change the text of the manuscript as follows:

The baseline Scr level was defined as the lowest Scr value that was available in the last 3 months prior to admission and throughout the hospital stay (page 7)

6.- The other observation is that patients with first admission were included. This creates a survival bias as patient may have died in subsequent admission

R= We understand the concern of the reviewer. 

We decided to consider only the first hospital admission since the nature of our study was mainly to compare HA-AKI vs CA-AKI. The event of CA-AKI would not be interpreted properly if the patient who had been discharged was hospitalized again, we think that this previous hospitalization strongly contributes to her readmission, making the conditions of CA-AKI completely different. Finally we believe that readmissions were <10%.

7.- The authors have used variable end-points. On some occasions, the authors report survival and on other occasion, mortality. The literature reports mortality for AKI. For example: The multivariable regression is using mortality as dependent variable which survival is quoted for HA & CA AKI

R=We appreciate the observation, we fully agree. We will change all the words surival for mortality when apropiate, you will find it on pages: 7, 8, 9, 11, 13, 14

8.- Was AKI not defined by KDIGO classification? This is not stated in methods

R= In methods section (page 6) you will find the phrase where we describe the definition of AKI by KDIGO.

AKI was diagnosed by the serum creatinine (Scr) KDIGO criteria (13)

---

## [Decision Letter · Decision Letter 1]

14 Jul 2021

PONE-D-21-11844R1

Mortality and evolution between Community and Hospital-acquired COVID-AKI

PLOS ONE

Dear Dr. Chávez-Iñiguez,

Thank you for submitting your manuscript to PLOS ONE. After careful consideration, we feel that it has merit but does not fully meet PLOS ONE’s publication criteria as it currently stands. Therefore, we invite you to submit a revised version of the manuscript that addresses the points raised during the review process.

We look forward to receiving your revised manuscript.

Kind regards,

Boris Bikbov

Academic Editor

PLOS ONE

Reviewers' comments:

Reviewer's Responses to Questions

**Comments to the Author**

1. If the authors have adequately addressed your comments raised in a previous round of review and you feel that this manuscript is now acceptable for publication, you may indicate that here to bypass the “Comments to the Author” section, enter your conflict of interest statement in the “Confidential to Editor” section, and submit your "Accept" recommendation.

Reviewer #1: (No Response)

Reviewer #2: (No Response)

2. Is the manuscript technically sound, and do the data support the conclusions?

Reviewer #1: Partly

Reviewer #2: Partly

3. Has the statistical analysis been performed appropriately and rigorously? 

Reviewer #1: Yes

Reviewer #2: Yes

4. Have the authors made all data underlying the findings in their manuscript fully available?

Reviewer #1: Yes

Reviewer #2: Yes

5. Is the manuscript presented in an intelligible fashion and written in standard English?

Reviewer #1: No

Reviewer #2: No

6. Review Comments to the Author

Reviewer #1: The authors have revised the document, but there remains inherent inconsistencies with the data.

Authors have 230 patients who did not have baseline within 3 months and were excluded. Did these patients have baseline creatinine in last 12 months? For other patients, baseline creatinine within a year was considered and there is no clear explanation offered by the authors.

The manuscript still has conflicting use of mortality and survival Eg: page 14

The study is retrospective observational study and consent was taken at the time of hospitalization. The conclusion states that this is prospective study

There are too many unnecessary tables. On page 14, please ensure confidence interval is correctly reported.

Kaplan Meier curve has time on x-axis, which does not make sense and should have days. Even with days, Kaplan Meier curve is inappropriate for short duration of follow-up of 28 days and does not add any information on long–term outcome

Reviewer #2: Dear authors,

The manuscript makes a different contribution to the understanding and management of patients with AKI-Covid. I highlight the data on the mortality score for AKI-Covid. However, the manuscript needs adjustments:

Abstract

-Use the KDIGO standardized nomenclature for renal replacement therapy-RRT (throughout the text) instead of KRT.

-Conclusions: add data on mechanical ventilation and LRA.

Introduction

-Subsidize with more references. Sugestions:

https://pubmed.ncbi.nlm.nih.gov/32416769

https://pubmed.ncbi.nlm.nih.gov/34033655

Methods

- I would like to see the ethics committee protocol number.

Results

-Format the tables. They are presented in frame format.

-I suggest unifying the results of tables 3 and 4; 5 and 6.

-Present the score data only with figure 4. Leave table 7 in the supplementary material.

- Figure 1 is incomplete. Please enter the subdivisions of patients who were elected in the study.

Discussion

-Increase the number of references.

-Better discuss the findings

For example: The Body Mass Index was high and although without statistical relevance in your study, it has been observed that overweight and obesity are risk factors for covid-19.

Att,

Reviewer

7. PLOS authors have the option to publish the peer review history of their article (what does this mean?). If published, this will include your full peer review and any attached files.

Reviewer #1: No

Reviewer #2: **Yes: **CASSIANE DEZOTI DA FONSECA

---

## [Author Response · Author response to Decision Letter 1]

1 Aug 2021

Authors response to the reviewers

We appreciate the excellent work of the reviewers, without a doubt their comments and opinions have been of great value to us, they add greater quality to the manuscript. We humbly hope that our responses will satisfy your requests.

Reviewer #1: The authors have revised the document, but there remains inherent inconsistencies with the data.

Authors have 230 patients who did not have baseline within 3 months and were excluded. Did these patients have baseline creatinine in last 12 months? For other patients, baseline creatinine within a year was considered and there is no clear explanation offered by the authors.

R= These 230 patients who were excluded was because lack of data prior to hospitalization, or who for some reason were not taken in hospitalization. We add in figure 1, point 5, the phrase lack of data.

The manuscript still has conflicting use of mortality and survival Eg: page 14

R=We changed the word mortality to survival in page 14.

Survival was used when we reported overall survival, the 28-day survival comparing CA-AKI, HA-AKI, and mechanical ventilation (Fig 2), and the 28-day survival between AKI stages and mechanical ventilation (S2 Fig).

Mortality was used when we reported the percentage of death among those with different types of AKI (Fig 3), and in the multivariable analysis of variables associated with death (Table 3).

The study is retrospective observational study and consent was taken at the time of hospitalization. The conclusion states that this is prospective study.

R=The data were recruited prospectively, from the hospitalization of the patients at the time of admission, informed consent was obtained for the storage of information. In the abstract and methods section, we change the word retrospective to prospective.

There are too many unnecessary tables. On page 14, please ensure confidence interval is correctly reported.

R= Done. We check the confidence intervals, and they are correct.

Kaplan Meier curve has time on x-axis, which does not make sense and should have days. Even with days, Kaplan Meier curve is inappropriate for short duration of follow-up of 28 days and does not add any information on long–term outcome.

R= The Kaplan-Meier procedure is a method of estimating time-to-event models in the presence of censored cases. The Kaplan-Meier model is based on estimating conditional probabilities at each time point when an event occurs and taking the product limit of those probabilities to estimate the survival rate at each point in time. 

We understand that the follow-up time is not very long compare to other long-term diseases but due to the severity of the disease that is being analyzed here, the use of a kaplan-meier analysis with a 28-day follow-up is adequate, not to mention that it also offers visual support for greater understanding of the reader. 

We also add a new supplemental material file with the figures change in DAYS instead of TIME.

Reviewer #2: Dear authors,

The manuscript makes a different contribution to the understanding and management of patients with AKI-Covid. 

I highlight the data on the mortality score for AKI-Covid. However, the manuscript needs adjustments:

Abstract

Use the KDIGO standardized nomenclature for renal replacement therapy-RRT (throughout the text) instead of KRT.

R= In the last KDIGO conference report 2020 it was attended to use “kidney” instead of “renal” or “nephro-“. They suggested to use “Kidney replacement therapy” and to avoid “Renal replacement therapy”. 

Kidney International (2020) 97, 1117–1129; https://doi.org/10.1016/ j.kint.2020.02.010

Conclusions: add data on mechanical ventilation and AKI.

Introduction

R= We add 6 more references about the strong association between mechanical ventilation and AKI, you can find them in page 20

Subsidize with more references. Sugestions:

https://pubmed.ncbi.nlm.nih.gov/32416769

https://pubmed.ncbi.nlm.nih.gov/34033655

R= We appreciate the suggestions of these 2 references, we include interesting information about them in the Discussion, in addition, we add 2 additional ones. You can find them with the number in pages 18, 19 and 20

Methods

I would like to see the ethics committee protocol number.

R= we add the committee protocol number in methods page 5.

The Institutional Review Board of both hospitals (Hospital Civil de Guadalajara HCG/CEI-0473/20 and Centro Medico Nacional 20 de Noviembre 09-240.2020)

Results

Format the tables. They are presented in frame format.

R= done, tables are presented in frame format.

I suggest unifying the results of tables 3 and 4; 5 and 6.

R= Done. Thanks for the suggestion, tables 3 and 4 ; 5 and 6 were unifying, in there would be only table 3 and 4. You can find them in pages 15, 16 and 17.

Present the score data only with figure 4. Leave table 7 in the supplementary material.

R= Done.

Figure 1 is incomplete. Please enter the subdivisions of patients who were elected in the study.

R= done, we included elected patients (in the first square)

Discussion

Increase the number of references.

R= We appreciate the suggestions. We found articles that we consider interesting to include in our manuscript, they were essentially added to the Discussion. You can find in page 17-21.

Better discuss the findings

For example: The Body Mass Index was high and although without statistical relevance in your study, it has been observed that overweight and obesity are risk factors for covid-19.

R=We understand the point of discussing the BMI as an interesting point, we appreciate that you have pointed it out. We add reference 25 (a meta-analysis on the relationship between obesity and COVID) and contrast our findings with this variable in the discussion, we believe that it certainly adds value to our manuscript.

---

## [Decision Letter · Decision Letter 2]

16 Aug 2021

PONE-D-21-11844R2

Mortality and evolution between Community and Hospital-acquired COVID-AKI

PLOS ONE

Dear Dr. Chávez-Iñiguez,

Thank you for submitting your manuscript to PLOS ONE. After careful consideration, we feel that it has merit but does not fully meet PLOS ONE’s publication criteria as it currently stands. Therefore, we invite you to submit a revised version of the manuscript that addresses the points raised during the review process.

There are a few items to correct, mainly considering the terminology. Particularly, this concerns the following:

- Please correct the titles of the tables "Univariable-Multivariable logistic regression..." to better readers' perception, and use the coherent terminology in the title and the column names (Univariate/Multivariate);

- Please exclude subtitles "-Mortality of CA-AKI vs HA-AKI: ", etc from the "Discussion", changing them with a short introductory phrase.

We look forward to receiving your revised manuscript.

Kind regards,

Boris Bikbov

Academic Editor

PLOS ONE

Journal Requirements:

Reviewers' comments:

Reviewer's Responses to Questions

**Comments to the Author**

1. If the authors have adequately addressed your comments raised in a previous round of review and you feel that this manuscript is now acceptable for publication, you may indicate that here to bypass the “Comments to the Author” section, enter your conflict of interest statement in the “Confidential to Editor” section, and submit your "Accept" recommendation.

Reviewer #2: All comments have been addressed

2. Is the manuscript technically sound, and do the data support the conclusions?

Reviewer #2: Yes

3. Has the statistical analysis been performed appropriately and rigorously? 

Reviewer #2: Yes

4. Have the authors made all data underlying the findings in their manuscript fully available?

Reviewer #2: Yes

5. Is the manuscript presented in an intelligible fashion and written in standard English?

Reviewer #2: Yes

6. Review Comments to the Author

Reviewer #2: Dear authors,

The manuscript is much better presented. My suggestions were successfully carried out. I consider the article suitable for publication.

7. PLOS authors have the option to publish the peer review history of their article (what does this mean?). If published, this will include your full peer review and any attached files.

Reviewer #2: **Yes: **cassiane dezoti da fonseca

---

## [Author Response · Author response to Decision Letter 2]

17 Aug 2021

Response to reviewers

We deeply appreciate the comments of the reviewers and are pleased that their requests have been successfully fulfilled.

We have made the 2 changes suggested by the editorial committee.

- Please correct the titles of the tables "Univariable-Multivariable logistic regression..." to better readers' perception, and use the coherent terminology in the title and the column names (Univariate/Multivariate)

R= Done, titles of Table 2 and 3 of the words "Univariable-Multivariable" by "Univariate / Multivariate"

-Please exclude subtitles "-Mortality of CA-AKI vs HA-AKI: ", etc from the "Discussion", changing them with a short introductory phrase.

R= Done, we exclude subtitles "-Mortality of CA-AKI vs HA-AKI:", "Risk factors for CA-AKI", "KRT between CA-AKI vs HA-AKI" and "Predictive score of death in patients with COVID-AKI" from the "Discussion". We believe that only by deleting those subtitles and leaving the first sentence of each paragraph as it is in the manuscript, the content is perfectly understood, without adding anything else.

---

## [Editor Report · Decision Letter 3]

20 Aug 2021

PONE-D-21-11844R3

Mortality and evolution between Community and Hospital-acquired COVID-AKI

PLOS ONE

Dear Dr. Chávez-Iñiguez,

Thank you for submitting your manuscript to PLOS ONE. After careful consideration, we feel that it has merit but does not fully meet PLOS ONE’s publication criteria as it currently stands. Therefore, we invite you to submit a revised version of the manuscript that addresses the points raised during the review process.

Thanks for correcting the minor items mentioned before. The changes you introduced are fine, but since after the acceptance there will be no additional editing, please perform further minor corrections to the tables' title to make it not only technically correct but also more easier to percept by readers who are primary clinicians. For example, the "Table 3. Univariable/Multivariable logistic regression model to determine the variables associated with mortality at 28-days follow up and Kidney replacement therapy in hospitalized COVID patients." could be changed to "Table 3. Factors associated with mortality at 28-days of follow up and the requirement of kidney replacement therapy in hospitalized COVID patients, in the univariable and multivariable logistic regression model." This is just an example, you could change the titles in your own way, but I hope this example demonstrates the idea. Please also change the terminology used in the table ("Grade COVID" -> "COVID severity", "Mechanical ventilator" -> "Need in mechanical lung ventilation") and also in the manuscript text ("ventilator" -> "ventilation", etc). You have prepared a nice manuscript with valuable results, there is only a need to polish a little bit the terminology.

We look forward to receiving your revised manuscript.

Kind regards,

Boris Bikbov

Academic Editor

PLOS ONE
---

## [Author Response · Author response to Decision Letter 3]

24 Aug 2021

Response to reviewers

We deeply appreciate the comments of the reviewers and are pleased that their requests have been successfully fulfilled.

We have welcomed comments from the Academic Editor Boris Bikbov about minor edits to the tables and text, which we have corrected according to their instructions. Now the manuscript is easier to read.

"Table 3. Univariable/Multivariable logistic regression model to determine the variables associated with mortality at 28-days follow up and Kidney replacement therapy in hospitalized COVID patients." could be changed to "Table 3. Factors associated with mortality at 28-days of follow up and the requirement of kidney replacement therapy in hospitalized COVID patients, in the univariable and multivariable logistic regression model." This is just an example, you could change the titles in your own way, but I hope this example demonstrates the idea. 

R=Done, The changes have been made in table 3 and 4 with the editor's suggestions, now it is as follows:

Table 3. Factors associated with mortality at 28-days of follow up and the requirement of kidney replacement therapy in hospitalized COVID patients, in the univariable and multivariable logistic regression model. (page 16)

Table 4. Factors associated with CA-AKI and HA-AKI in hospitalized COVID patients, in the univariable and multivariable logistic regression model. (page 17)

Please also change the terminology used in the table ("Grade COVID" -> "COVID severity", "Mechanical ventilator" -> "Need in mechanical lung ventilation") and also in the manuscript text ("ventilator" -> "ventilation", etc). 

R= Done, we have made the changes suggested by the Editor in tables and the manuscript, of the terms: grade COVID, mechanical ventilator; by the terms: COVID severity, Need in mechanical lung ventilation.

We really hope that this changes satisfy to the Academic Editor.

---

## [Editor Report · Decision Letter 4]

6 Sep 2021

Mortality and evolution between Community and Hospital-acquired COVID-AKI

PONE-D-21-11844R4

Dear Dr. Chávez-Iñiguez,

We’re pleased to inform you that your manuscript has been judged scientifically suitable for publication and will be formally accepted for publication once it meets all outstanding technical requirements.

Kind regards,

Boris Bikbov

Academic Editor

PLOS ONE

Additional Editor Comments (optional):

ACADEMIC EDITOR: Please note there are some terminology and formatting incongruences in the tables. For example, the tables' title indicates "... in the univariable and multivariable logistic regression model" that is fine. However, the authors have left the column names as "Univariate (95% CI)" and "Multivariate (95% CI)" - that is not correct and should be "Univariable analysis, OR (95% CI)". In another table the authors compared 2 groups and named the column with p-value as "P value (CA-AKI vs. HA-AKI)", with excessive detail "(CA-AKI vs. HA-AKI)" because that are the groups' names.

The journal office will include the requested edits as part of the final requirements the authors must complete before editorial acceptance. Please correct these formatting incongruencies before the manuscript can be editorially accepted.
---

## [Editor Report · Acceptance letter]

26 Oct 2021

PONE-D-21-11844R4 

Mortality and evolution between Community and Hospital-acquired COVID-AKI 

Dear Dr. Chávez-Iñiguez:

I'm pleased to inform you that your manuscript has been deemed suitable for publication in PLOS ONE. Congratulations! Your manuscript is now with our production department. 

Kind regards, 

on behalf of

Dr. Boris Bikbov 

Academic Editor

PLOS ONE